# Tetraploidization Increases the Motility and Invasiveness of Cancer Cells

**DOI:** 10.3390/ijms241813926

**Published:** 2023-09-10

**Authors:** Mohamed Jemaà, Renee Daams, Slim Charfi, Fredrik Mertens, Stephan M. Huber, Ramin Massoumi

**Affiliations:** 1Department of Laboratory Medicine, Translational Cancer Research, Faculty of Medicine, Lund University, 22381 Lund, Sweden; renee.daams@med.lu.se; 2Human Genetics Laboratory (LR99ES10), Faculty of Medicine of Tunis (FMT), Tunis El Manar University, Tunis 1006, Tunisia; 3Department of Biology, Faculty of Science of Tunis, Tunis El Manar University, Tunis 2092, Tunisia; 4Department of Pathology, Habib Bourguiba Hospital, Sfax University, Sfax 3029, Tunisia; charfislim@gmail.com; 5Department of Laboratory Medicine, Division of Clinical Genetics Lund University, 22381 Lund, Sweden; fredrik.mertens@med.lu.se; 6Department of Radiation Oncology, University Hospital of Tübingen, Hoppe-Seyler-Str. 3, 72076 Tübingen, Germany; stephan.huber@uni-tuebingen.de

**Keywords:** colon cancer, sarcoma, tetraploidy, migration, invasion, metastasis

## Abstract

Polyploidy and metastasis are associated with a low probability of disease-free survival in cancer patients. Polyploid cells are known to facilitate tumorigenesis. However, few data associate polyploidization with metastasis. Here, by generating and using diploid (2n) and tetraploid (4n) clones from malignant fibrous histiocytoma (MFH) and colon carcinoma (RKO), we demonstrate the migration and invasion advantage of tetraploid cells in vitro using several assays, including the wound healing, the OrisTM two-dimensional cell migration, single-cell migration tracking by video microscopy, the Boyden chamber, and the xCELLigence RTCA real-time cell migration. Motility advantage was observed despite tetraploid cell proliferation weakness. We could also demonstrate preferential metastatic potential in vivo for the tetraploid clone using the tail vein injection in mice and tracking metastatic tumors in the lung. Using the Mitelman Database of Chromosome Aberrations in Cancer, we found an accumulation of polyploid karyotypes in metastatic tumors compared to primary ones. This work reveals the clinical relevance of the polyploid subpopulation and the strategic need to highlight polyploidy in preclinical studies as a therapeutic target for metastasis.

## 1. Introduction

Metastasis is defined as the spread of cancer cells from the primary tumor site to surrounding tissues and to distant organs [1]. Metastasis is executed via a complex process called metastasis cascade. It occurs through progressive steps starting from the invasion of adjacent tissues, intravasation, transport via the circulatory system, arrest at a secondary site, extravasation, and growth in a secondary organ [2].

Despite the great improvements in cancer diagnosis and therapy, tumor invasion and metastasis are the main causes of tumor recurrence and patient morbidity, causing 90% of human cancer deaths [3]. Metastasis is by far the least understood aspect of cancer, and the role of its genetic and biochemical determinants remains poorly studied.

The diploid karyotype is the state of having two complete sets of homologous chromosomes (2n). Polyploidization is the increase in genome size caused by the inheritance of an additional set of chromosomes. One of the most common polyploid stages is tetraploidization (4n) [4]. Some tolerable physiological conditions of polyploid and tetraploid cells exist in the organism, such as hepatocytes, syncytiotrophoblasts, megakaryocytes, and myocytes [5]. However, illicit cell polyploidization has been associated with human diseases, including cancer, by enhancing mitotic dysfunctions and genomic and chromosomal instability. Three processes may induce tetraploidy, namely cell fusion, endoreplication (1 single nucleus due to karyokinesis failure), or endomitosis (2 nuclei due to cytokinesis failure) [4].

Tetraploidy contributes to oncogenesis as they constantly undergo chromosomal rearrangements when cycling. This chromosomal instability provokes aneuploidy by multipolar divisions and/or illicit bipolar divisions in the presence of incorrect microtubule-kinetochore attachments. The resulting viable aneuploid cells drive tumorigenesis as they are resistant to apoptosis and show enhanced proliferation and metastasis [4,6,7].

Tetraploid subpopulation has been detected at early stages of multiple cancer cell types (including bronchial, esophageal, gastric, mammary, colorectal, ovarian, cervical, and prostate carcinomas) [5]. Tetraploidy is correlated with the inactivation of the tumor suppressors retinoblastoma 1 (RB1) and tumor protein p53 (TP53). Moreover, the inactivation of p53 facilitates the tetraploidization of cell lines [8,9,10] and primary epithelial cells from the colon and the mammary gland [11,12].

Interestingly, several studies showed that metastatic tumors contain considerable proportions of polyploid cells compared to primary tumors [8,13,14,15,16,17,18]. These reports suggest a certain role of polyploidy in improving metastasis; however, few data exist concerning the potential involvement of tetraploidization in promoting cell migration and/ or invasion. 

The aim of this study is to investigate the motility and invasion of diploid vs.. tetraploid cancer cells using two different cancer types, namely colon carcinoma and sarcoma.

## 2. Results

### 2.1. Generation and Purification of Tetraploid Clones

To study the behavior of tetraploid cells in terms of migration and invasion, we created diploid and tetraploid clones. To avoid tissue specificity, we used 2 types of cancer cells, namely human colon carcinoma RKO and malignant fibrous histiocytoma MFH152 (soft tissue sarcoma subtype). We generated a series of tetraploid and diploid clones from MFH152 via flow cytometry using limiting dilution sub-cloning. Indeed, We showed in a previous study that these cells are heterogeneous and contain tetraploid subpopulations [19]. We assumed that tetraploid cells are bigger than diploid cells and sorted clones, based on size and granularity parameters, using the normal light scattering parameters forward scatter (FSC) vs. side scatter (SSC) gating (Appendix A). These clones were grown in 96-well plate wells and transferred to 48-, 24-, 12-, and 6-well plates to obtain established cells. The first passage from a 6-well plate to a 10 cm petri dish was considered passage 1. Flow cytometry analyses at different cell passages (from 1 to 15) confirmed the tetraploid and diploid status of the obtained clones. We decided to use early passage clones to ensure their genomic stability.

For the human colon carcinoma RKO cells, we used another strategy. Cells were transiently (2 days) exposed to 600 ng/mL of cytochalasin D, a reversible inhibitor of the actin cytoskeleton that blocks cytokinesis. In these conditions, tetraploidization was induced, and several tetraploid clones survived. Diploid cells in the G2/M phase have the same DNA amount as tetraploid cells in G1 (both contain a 4n-equivalent DNA content). To avoid any contamination between the 2 karyotypes of clones, we treated cells with a low dose of Hoechst 33342 as a live and reversible dye for the cell cycle and sorted diploid cells in G1 phase (2n) and tetraploid cells in G2/M phase (8n-equivalent) (Appendix A). Clones were grown in a 96-well plate and followed until the establishment of cells, as mentioned before. In total, for this study, we used 2 diploid and 2 tetraploid RKO clones and 3 diploid and 4 tetraploid MFH152 clones.

### 2.2. Characterization of Diploid and Tetraploid Clones

We evaluated the first clone’s size using flow cytometry and light scattering parameters forward scatter (FSC). Here, we found that tetraploid clones are slightly bigger than diploid clones, both from RKO and MFH152 origins (Figure 1A,B, and Appendix A). Light microscopy analysis confirmed this observation. Cells were fixed and stained with phalloidin and DAPI to observe the cytoskeleton architecture and the nucleus of the clones (Figure 1C). Immunofluorescence further confirmed that tetraploid clones are bigger than diploid ones. We quantified the nucleus area using the Image J software (V3.8, https://imagej.nih.gov/ij/), and found that both RKO and MFH152 tetraploid clones have a bigger nucleus than diploid clones (Figure 1D and Appendix A). The metaphase spread experiment showed that diploid clones have around 46 chromosomes while tetraploid ones have double (Figure 1E,F and Appendix A). Ultimately, we show here a cell cycle analysis as a supplementary indication of the ploidy status of the different clones (Figure 1G and Appendix A).

To study proliferation dynamics of the diploid and tetraploid clones, we quantified the different phases of the clone’s cell cycle. In tetraploid clones, we found a significant accumulation in the G1 phase compared to diploid clones. This suggests that tetraploid clones may proliferate less than their diploid counterpart (Figure 1H and Appendix A). To confirm this observation, we performed a crystal violet proliferation assay and discovered that both RKO and MFH152 tetraploid clones were less proliferative than diploids (Figure 2A,B).

### 2.3. Tetraploid Clones Are More Motile and Invasive Than Diploid In Vitro

Our previous data showed that sarcoma tetraploid clones were more motile than diploid cells in vitro [19]. We decided to compare the motile properties of diploid and tetraploid RKO colon carcinoma clones. Motility was first assessed using a wound-healing assay. For up to 48 h, wound closure was compared between diploid and tetraploid clones. The decrease of the cell-free area was significantly enhanced with tetraploid clones comparatively to diploid ones, both at 24 h and 48 h (Figure 3A). In a second experimental approach, migration was quantified using a two-dimensional assay from Oris^TM^. The assay comprised confluent cells seeded in a 96-well plate with a silicon insert. When cell seeding stoppers were removed, the cell-free area or migration zone was measured for up to 48 h. The cell-free area was smaller in tetraploid wells than in diploid ones, confirming the enhanced migration of tetraploid cells (Figure 3B). Cell/cell junctions play important functions during collective migration; thus, migrating cells may behave differently during individual migration. Using time-lapse microscopy, we followed the individual cell motility of diploid and tetraploid RKO clones for 24 h. As for collective cell migration, we found that individualized tetraploid cells were significantly more motile than diploid cells (Figure 3C). To assess the invasion potential of the cells, we performed a Boyden chamber assay. We found that tetraploid clones were more invasive than diploid ones over 24 h (Figure 4A). Moreover, we used the real-time 3D invasion assay xCELLigence technology and found that tetraploid clone enhances invasion for up to 24 h comparatively to diploid RKO clone (Figure 4B). In conclusion, we showed that tetraploid clones are more motile and invasive in vitro than diploid clones.

### 2.4. Tetraploid Clones Are More Metastatic Than Diploid In Vivo

To evaluate the possible advantage of tetraploid cells in vivo compared to diploid ones, we performed a tail vein assay. 12 NSG mice were injected intravenously (tail vein) with one million diploid or tetraploid MFH152 clones/mice, respectively, with six mice per group. Mice were sacrificed after eight weeks to assess cell infiltration in the lung as the target organ. Lungs were collected, fixed, and embedded prior to HE staining. We found more suspicious lung lesions in mice injected with MFH152 tetraploid clones than those injected with diploid clones. In four out of six mice injected with the tetraploid clone, we found an area of cellular proliferation in the lung parenchyma and dense cellular infiltration of the tissue with cellular atypia. A similar lesion was found in only one mouse of six injected with diploid clones (Figure 5A,B). This finding confirms the preferential metastatic effect of tetraploid cells in vivo.

### 2.5. Accumulation of Polyploid Cells in Metastatic Sarcoma Sites

First, to determine whether polyploidy and chromosomal instability (CIN) are associated with human metastases, we took advantage of the Mitelman Database of Chromosome Aberrations in Cancer (Mitelman, F., Johansson, B. & Mertens, F. Mitelman Database of Chromosome Aberrations and Gene Fusions in Cancer https://cgap.nci.nih.gov/Chromosomes/Mitelman (2019)) and we analyzed the karyotype of 63 matched pairs of primary vs. metastatic sarcoma tumors. Metastases showed enrichment for cells with near-triploid (3n) and near-tetraploid (4n) karyotypes comparatively to primary tumors that showed a predilection for near-diploid (2n) karyotypes (Figure 5C). Moreover, this tendency was even accentuated when comparing a primary tumor with the first and second sites of metastasis (Figure 5D). This finding further confirms the correlation between polyploidy and tumor metastasis.

## 3. Discussion

Metastasis is the leading cause of cancer mortality, causing up to 90% of human cancer deaths. Previous studies introduced the correlation between chromosomal instability, polyploidy, and tumor metastasis. In this study, we confirmed the preferential migration of polyploid cancer cells using in vitro and in vivo models and patient data extracted from the Mitelman Database of Chromosome Aberrations.

We first generated and characterized stable tetraploid and diploid clones from colon cancer carcinoma RKO and soft tissue sarcoma MFH152. We used these clones to investigate the cell’s migration, invasion, and metastasis using several established assays. We confirmed the strong correlation that exists between polyploidy and cell migration.

Polyploidy is a non-tolerated physiological state in proliferating cells. Polyploid cells are usually eliminated by apoptosis after the tetraploid G1 checkpoint [20,21] or by immunosurveillance mechanism [22]. Thus, the polyploidization process is usually found during conditions of stress, aging, and disease, especially cancer [23]. In the tumorigenesis context, the illicit survival of polyploid cells is correlated with the deficiency of the p53 and Rb pathways, as they are required for the tetraploidy checkpoint [24,25,26].

Polyploidy, the state of having more than a double set of chromosomes, is a metastable intermediate between diploidy and aneuploidy and a promoter of chromosomal and genomic instability [4]. Several types of machinery can provoke this mechanism, named the polyploidization/depolyploidization cascade. The most instinctive one is mitosis deregulation due to the increase in chromosome number and the presence of supernumerary centrosomes. This leads to a multipolar mitotic spindle and possibly to a multipolar division and the generation of aneuploid cells [4,9]. Survived daughter cells play an essential role in tumor development, and several studies documented the implication of genomic instability in the oncogenesis process [27,28].

In our study, we introduced a unique role of polyploid cancer cells in promoting metastasis. Indeed, despite a proliferative disadvantage, tetraploid cells showed increased migratory and invasive capacities.

Recently, several studies correlated metastasis and cell ploidy with metastatic tumors containing considerable proportions of polyploid and chromosomally unstable cells compared to primary tumors. We can exemplify several types of cancer, including renal cell carcinoma, melanoma, sarcoma, non-small cell lung, pancreatic, prostate, breast, ovarian, thyroid, and salivary gland metastases [8,13,14,29,30,31]. In addition, a previous study showed a preferential metastatic potential of polyploid cells using an in vivo mice model and melanoma mousse cell line B16 diploid vs. tetraploid clones [32]. Moreover, a recent in vitro experimentation discussed the preferential migration potential of polyploid clones comparatively to diploid ones [33] in addition to our previous study using malignant fibrous histiocytoma clones [19].

The mechanisms by which polyploid cells may acquire preferential and enhanced migration and invasive capacity are poorly investigated. However, some interesting hypothesis has been proposed. The presence of extra centrosomes could promote invasiveness in cancer cells [34,35]. Moreover, some polyploid cells showed an improved acquisition of epithelial to mesenchymal transition EMT [36]. It has also been shown that polyploidy is associated with the main transcriptional regulators of EMT, including SNAI, TWIST, and N-cadherin [37]. Other work presented an up-regulation of certain genes related to the invasive/migratory phenotype [38], while other studies displayed a metabolic reprogramming property [39]. A recent report has shown that polyploid cancer cells exhibit a downregulation of genes associated with the cell membrane. This may help them more easily detach from the extracellular matrix and adjacent cells and de facto enhance their motility and migration [40].

In addition to the internal biological properties of polyploid cells, intriguing studies have shown that polyploidy is required to maintain cells in a pro-metastatic state by activating the internal immune system and mimicking chronic inflammation, which helps cells spread [8,41]

The major impact of this study opens new windows regarding cancer therapy through introducing the migratory and metastatic advantage of polyploid cancer cells. In the future, additional investigations will be engaged to study this preferential molecular mechanism.

## 4. Materials and Methods

### 4.1. Cell Lines, Culture Conditions and Reagents

Human colon carcinoma RKO clones and parental cell lines were grown in McCoy’s 5A medium supplemented with 10% fetal calf serum (FCS), 10 mM HEPES buffer, 1 mM sodium pyruvate buffer, and antibiotics. MFH152 clones and parental cell lines were grown in Dulbecco’s modified Eagle’s medium (DMEM) supplemented with 10% FCS and antibiotics. Cells were routinely maintained at 37 °C under 5% CO_2_. Cells were seeded onto the appropriate supports (6-, 12-, 24-, or 96-well plates, 100 mm Ø Petri dishes) 24 h before the beginning of the experiment. Cytochalasin D and Nocodazole were stocked as 10 mM solution in DMSO. All the material was purchased from (Sigma-Aldrich, Stockholm, Sweden)

### 4.2. Cytofluorometric Sorting 

#### 4.2.1. I/Sorting Based on Cell Size

Cytofluorometric sorting of diploid and tetraploid clones was performed with a FACSAria cell sorter, and the gating of small and big cells was based on the size and granularity parameters using the normal light scattering parameters forward scatter (FSC) vs. side scatter (SSC). Single cells, sorted from the MFH152 mother cell line, were seeded in 96-well plates. 1000 “small” plus 1000 “big” clones were sorted. After 15 days, surviving clones were cultured in 6-well plates. We succeeded in isolating stable diploid and tetraploid clones. We should note that the majority of surviving clones were aneuploid. 

#### 4.2.2. II/Sorting Based on Cell Cycle

RKO mother cell line was treated for 48 h with 600 ng/mL of cytochalasin D before a washout and culture. Cells were then treated with 1 μg/mL Hoechst 33,342, and cytofluorometric sorting of diploid and tetraploid clones was performed with a FACSAria cell sorter. Gating of G1 diploid vs. G2/M tetraploid was used to sort tetraploid and diploid clones. Clones were collected in 96-well plates and grown until final culture in petri dishes. We succeeded in isolating stable diploid and tetraploid clones.

### 4.3. Cell Cycle Analysis

For the assessment of cell cycle distribution, cells were collected, washed once with PBS, and then fixed by gentle vortexing in ice-cold 75% (*v/v*) ethanol for 30 s. After overnight incubation at −20 °C, samples were centrifuged, PBS washed, and stained with 50 μg/mL PI in 0.1% (*w/v*) D-glucose in PBS supplemented with 1 μg/mL (*w/v*) RNase A (Sigma–Aldrich) for 30 min at 37 °C. Afterward, samples were incubated overnight at 4 °C before cytofluorometric analysis.

### 4.4. Quantification of Cell Size

Cells were collected and acquired using flow cytometry and the normal light scattering parameters of forward scatter (FSC) vs. side scatter (SSC). The histogram of forward scatter (FSC) was set to a linear scale. The threshold of FSC was set at the default value of “300” using Geo Mean.

### 4.5. Immunofluorescence Microscopy

Cells were cultured on coverslips in 6-well plates for 24 h and then fixed in 100% methanol for 10 min at −20 °C. Next, the cells were incubated in 10% FBS-PBS for 1 h at room temperature with phalloidin-Atto647 for the actin staining and DAPI for DNA staining (Sigma-Aldrich). Coverslips were then washed and mounted in a fluorescence mounting medium (Dako, Santa Clara, CA, USA). The images were obtained using a Zeiss LSM710 confocal microscope.

### 4.6. Nucleus Area Quantification

Stack images for immunofluorescence were acquired and analyzed with Image J software (V3.8, https://imagej.nih.gov/ij/). Nucleus areas were quantified using the object Intensity Segmentation Threshold.

### 4.7. Chromosome Spreads

Cells were treated with 100 nM nocodazole for 16 h to enrich the percentage of the mitotic population, then collected and subjected to hypotonic lysis by incubation in 75 mM KCl for 10 min at 37 °C. After removing the hypotonic solution, cells were fixed in freshly prepared Carnoy solution (3/1 methanol/acetic acid) and stored at –20 °C. Fixed cells were dropped onto pre-cooled glass microscope slides and dried at room temperature. Chromosomes were stained with 100 ng/mL DAPI and mounted in a fluorescence mounting medium (Dako, Santa Clara, CA, USA). The images were obtained using a Zeiss LSM710 confocal microscope.

### 4.8. Crystal Violet Proliferation Assay

Cells were seeded in 96-well plates with a 2000 cells/well density and cultured for up to 5 days. At every time point, cells were washed once in PBS and fixed with 4% paraformaldehyde (PFA) for 15 min. The PFA was removed, and cells were stained for 30 min at room temperature with an aqueous solution containing 0.1% (*w/v*) crystal violet. Cells were washed three times with distilled water before administering 200 µL/well of 10% acetic acid and shaking with micropipettes. The absorbance of each sample was measured using a scanning microplate spectrophotometer reader (Synergy 2, Biotek, Germany) by absorbance detection at 595 nm.

### 4.9. Migration and Invasion Assays

(1) For the wound-healing assay, scratches were performed on confluent cell monolayers with sterile 200 μL tips and monitored for up to 48 h by microscopy. Migration distances were expressed as percentages over control values. Indeed, as the scratches are not equal at time 0, we normalized the values by dividing every value by the time 0 value in every condition (Remaining free area /free area time 0).

(2) The two-dimensional Oris^TM^ cell migration assays were performed according to the manufacturer’s instructions (Platypus Technologies, Madison, WI, USA). Briefly, cells were seeded (4 × 10^4^ cells per well) into 96-well plates with a “silicone stopper” and grown overnight. Then, the stoppers were removed, and the cells were incubated for an additional 48 h to allow their migration into the empty zone. Data acquisitions were performed using the Axiovert 200 M Zeiss microscope. Migration distances were expressed as remaining cell-free areas, and we normalized the value to time 0 (Remaining free area /free area time 0).

(3) For individual cell migration, cells were seeded at low concentrations to avoid cell-cell contacts that would affect analyses and monitored by time-lapse microscopy. Image acquisitions were performed every hour for 24 h using an Inverted Axio Observer Z1 microscope equipped with ZEN2010 software (ZenBlue, Zeiss). Time lapse were analysed using ImageJ software (V3.8, https://imagej.nih.gov/ij/).

(4) For the Boyden chamber assay (or transwell assay), 1,000,000 RKO cells or 500,000 MFH152 cells were added to the upper chamber in serum-free media, and migration at 37 °C towards 10% FBS containing growth media was determined 24 h. Cells were briefly fixed with 4% PFA for 15 min and then stained for 30 min at room temperature with an aqueous solution containing 0.1% (*w/v*) crystal violet. Cells were washed three times with distilled water. Pictures were taken using an Axiovert 200 M Zeiss microscope, and the number of migrating cells was counted.

(5) Real-time cell migration measurements were performed using the xCELLigence RTCA technology. For these experiments, we used CIM-16-well plates, which have interdigitated gold microelectrodes on the underside of a filter membrane positioned between a lower and an upper chamber. The lower chamber was filled with a complete medium supplemented with 10% serum (acting as a chemoattractant). Cells (2 × 10^4^ cells/well) were seeded on top in a serum-free medium. Microelectrodes detect impedance changes that are proportional to the number of migrating cells and are expressed as cell index. Migration was monitored in real time for 24 h.

### 4.10. In Vivo Metastasis Assay

12 NSG mice were purchased from Taconic Laboratory (Ejby, Denmark). The animal experiments were performed according to the national and international guidelines of the European Union. Moreover, the protocol for the in vivo metastasis assay on mice was approved by the Center of Ethical Committee on Animal Experiments in Sweden “Centrala försöksdjurnämnden”. Ethical Dnr: M129-15. All mice were housed under pathogen-free conditions in the animal facility and received autoclaved water and food. Eight-week-old NSG mice were used in the study. Diploid or tetraploid MFH152 cells were suspended in phosphate-buffered saline (PBS). A total of 1 × 10^6^ cells/mouse (100 μL) were injected i.v. (intra-veinous injection); 6 mice for each group. Mice were sacrificed after 8 weeks to evaluate cell infiltration in the lungs.

### 4.11. Immunohistochemistry Staining

Formalin-fixed, paraffin-embedded lung sections (5 µm) were deparaffinized using routine techniques and placed in 200 mL of EnVisionTM target retrieval solution (pH 6.0; Dako, Hamburg Germany) for 20 min at 100 °C. After cooling for 20 min, slides were quenched with 3% H_2_O_2_ for 5 min. Immunostaining was visualized using the EnVisonTM + kit (Dako). In addition, slides were also stained with hematoxylin and eosin.

### 4.12. Statistical Procedures

Data are expressed as arithmetic means ± SEM. Statistical analysis was made with a GraphPad Prism using ANOVA with Tukey’s test as post-hoc.

Karyotype analysis was evaluated using the Fisher test of variances.

## Figures and Tables

**Figure 1 ijms-24-13926-f001:**
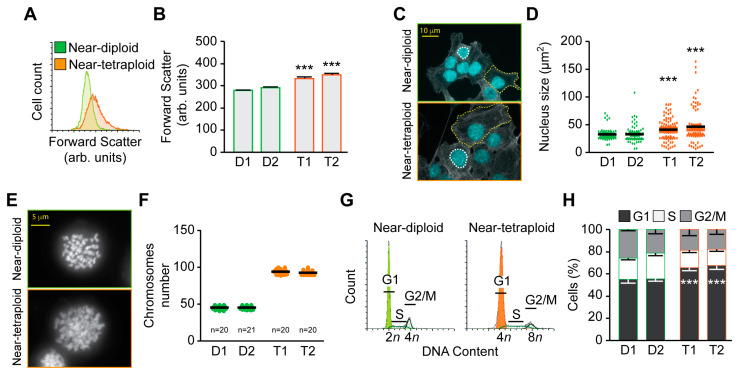
Diploid vs. tetraploid clones characterization. (**A**,**B**) Cell size comparison using flow cytometry and light scattering parameters. (**A**) shows original histograms of RKO diploid and tetraploid clones forward scatter (labeled in green and orange, respectively), and quantitative data are displayed in (**B**). (**C**,**D**): Cell shape analysis and nucleus area analysis. Representative microphotographs of diploid and tetraploid RKO clones labeled with actin (Phalloidin) and DNA (DAPI) staining are shown in (**C**). 1 Nucleus is surrounded by a white line while 1 cell is surrounded by a yellow line for each condition. Scale bar = 10 μm. Quantitative data of the nucleus area are displayed in (**D**). (**E**,**F**): Chromosome number counts. Metaphase spread of diploid and tetraploid RKO clones was performed, and representative microphotographs of DAPI-stained chromosomes are shown in (**E**). Respective quantitative data are displayed in (**F**). (**G**,**H**): Cell cycle analysis. Cell cycle distribution was assessed by flow cytometry. Diploid and tetraploid RKO clones were collected and stained with propidium iodide. Representative cell cycle histograms are shown in (**G**), and quantitative data are reported in (**H**). D refers to diploid, and T to tetraploid clone. Data in (**D**,**F**) are reported as individual values and mean and in (**B**,**H**) as means ± SEM; *n* = 5. *** (*p* < 0.001) indicates a significant difference between every tetraploid clone compared to diploid clone D1 (using the ANOVA test).

**Figure 2 ijms-24-13926-f002:**
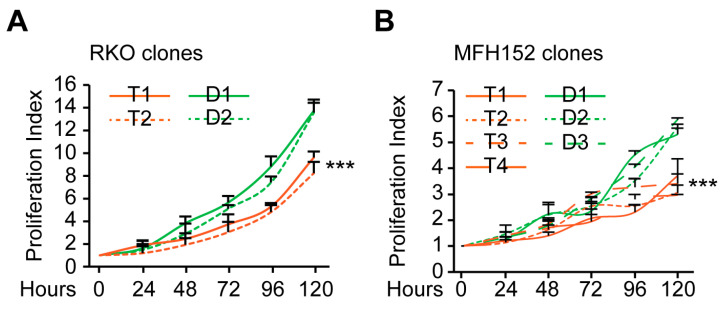
Evaluation of Diploid vs. tetraploid clones proliferation. (**A**,**B**): Cell proliferation. RKO (**A**) and MFH152 (**B**) diploid and tetraploid clones (labeled in green and orange, respectively) were cultured for 5 days, and proliferation was assessed using a crystal violet assay. The proliferative index is shown. D refers to diploid and T to tetraploid clone. Data are reported as means + SEM; *n* = 5. *** (*p* < 0.001) indicates a significant difference between every tetraploid clone compared to diploid clone D1 (using the ANOVA test).

**Figure 3 ijms-24-13926-f003:**
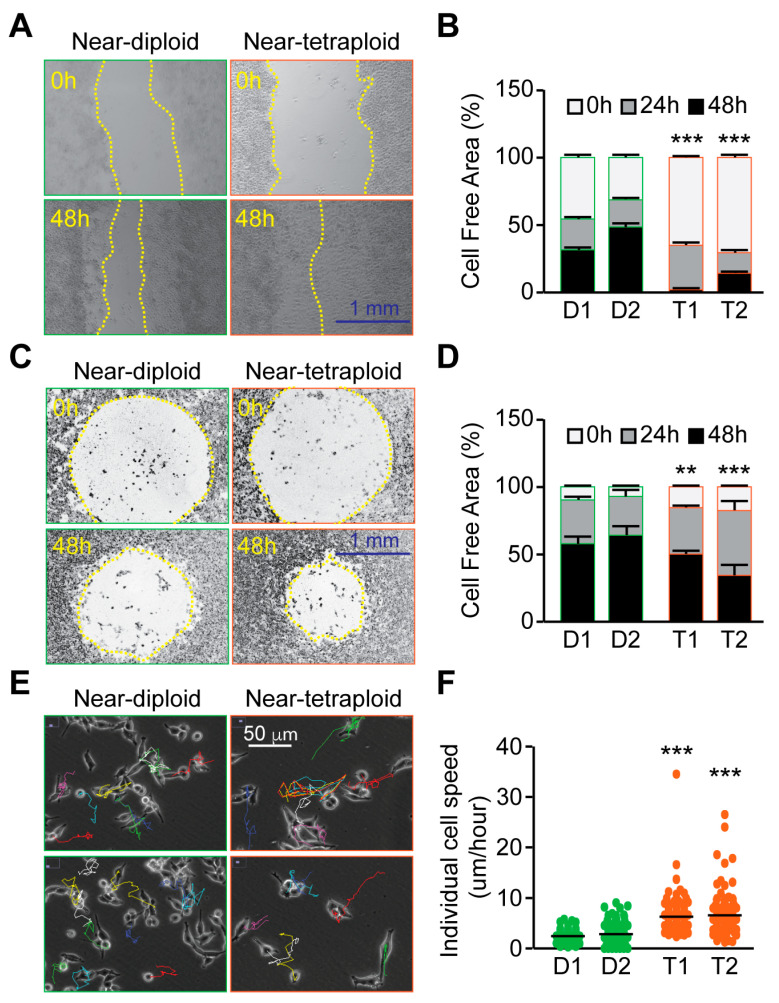
Tetraploid Clones are more motile than diploid clones. (**A**,**B**): Confluent diploid and tetraploid human colon carcinoma RKO cell monolayers were scratched with pipet tips and imaged for up to 48 h to study collective migration. Representative microphotographs are shown. The yellow broken lines delimit the cell-free area. A quantitative histogram shows the average cell-free area at 24 h and 48 h calculated from the wound closure rate (normalized to 0 h) using image J software (V3.8, https://imagej.nih.gov/ij/). (**C**,**D**): Two-dimensional migration assay using the Oris™ cell assay. Cells were allowed to migrate for up to 48 h after removing cell seeding stoppers to evaluate their motile potential. Representative photomicrographs are shown. The yellow dashed lines show the empty migration zone. Quantitative data are presented for time 24 h and 48 h. (**E**,**F**): diploid and tetraploid RKO clones were grown in non-confluent conditions and imaged for 24 h, using time-lapse microscopy, to evaluate the respective migration potential of individual cells. The panel shows the representative micrographs of the trajectories of some cells reconstituted using image J software (V3.8, https://imagej.nih.gov/ij/). D refers to diploid and T to tetraploid clones. Quantitative data shows the individual cell’s speed. Data are reported as SEM; *n* = 5. ** (*p* < 0.01) and *** (*p* < 0.001) indicate significant differences between every tetraploid clone compared to diploid clone D1 (using the ANOVA test).

**Figure 4 ijms-24-13926-f004:**
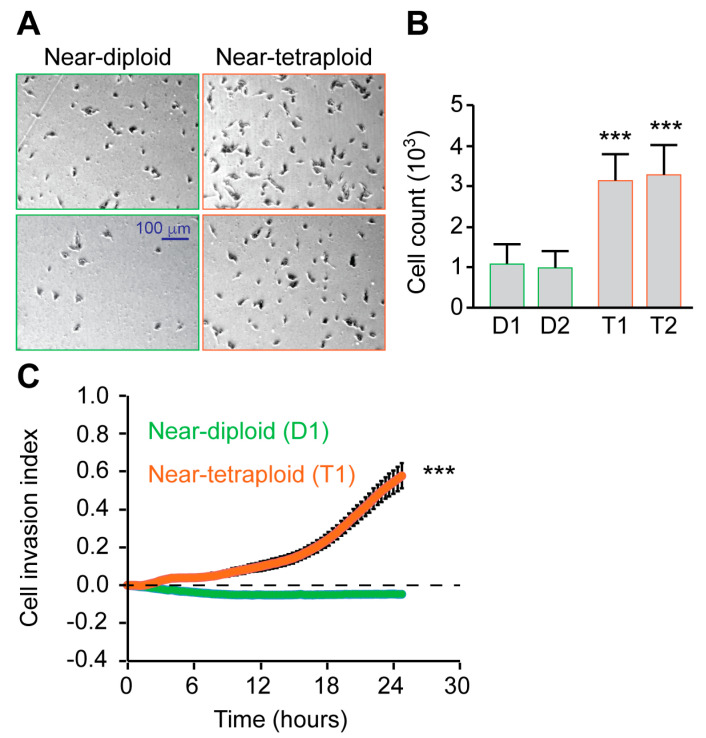
Tetraploid clones are more invasive than diploid clones. (**A**,**B**): Boyden chamber assay. Diploid and Tetraploid clones were collected, washed, and suspended in a free serum medium. Cells were then added to the upper compartments of the Boyden chamber and cultured for 24 h. Representative photomicrographs of cells that migrated 24 h later to the lower side of the filter are shown. Quantitative data of migrated cell numbers are presented on the right of the panel. (**C**): xCELLigence invasion assay. Diploid and tetraploid clones were washed, collected, and suspended in a free serum medium. Cells were then added to the upper compartments of the CIM-Plate for real-time impedance recording. Mean impedances of the cells are measured for 26 h. D refers to diploid and T to tetraploid clones, respectively. Data are reported as means ± SEM; *n* = 3. *** (*p* < 0.001) indicate significant differences between every tetraploid clone compared to diploid clone D1 (using ANOVA test).

**Figure 5 ijms-24-13926-f005:**
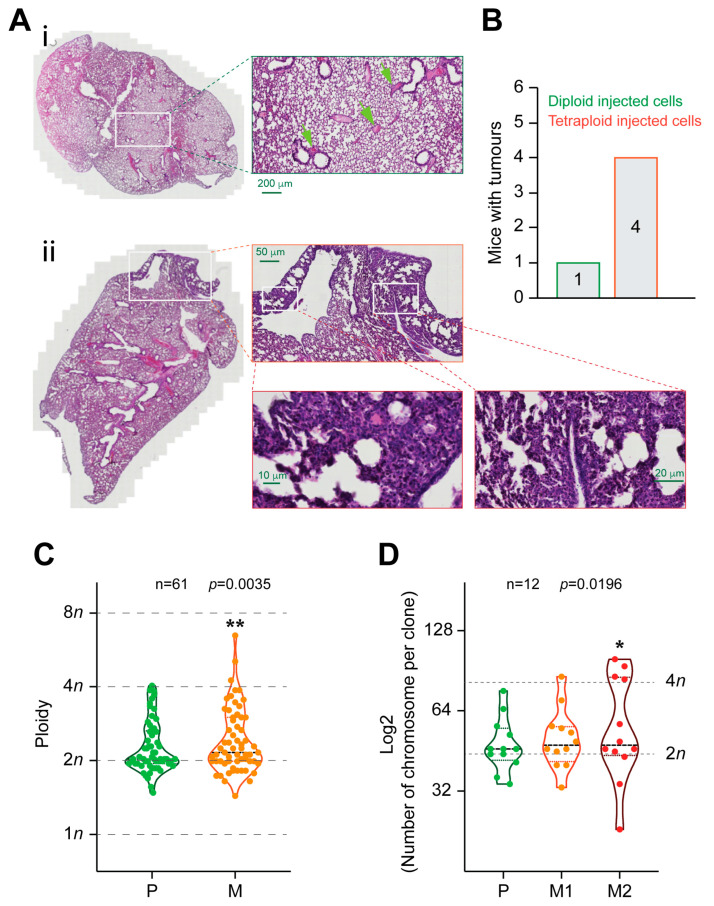
(**A**,**B**). Metastatic advantage of tetraploid sarcoma clone in vivo.12 NSG mice were injected intravenously in the tail vein with diploid or tetraploid MFH152 clones (6 mice/group). Mice were sacrificed after eight weeks to evaluate cell infiltration in the lungs. (**A**): i represents a lung section of a mouse injected with diploid cells. At high magnification, we can see the normal morphology of lung tissue. Green arrows indicate congestive vessels. (**A**): ii represents a lung section of a mouse injected with tetraploid cells. We can see an area of cellular proliferation in the lung parenchyma. At high magnification, we can see a dense cellular infiltration of the tissue with cellular atypia. (**B**) represents the number of mice with cancer cell infiltration. (**C**,**D**): Sarcoma metastases enrich for polyploidy. (**C**): Karyotype probability density between primary and metastatic tumors, *n* = 61. (**D**): Karyotype probability density between primary and metastatic tumor sites 1 and 2, *n* = 12. Data in (**C**,**D**) are reported as violin plots. * (*p* < 0.05) and ** (*p* < 0.01) indicate a significant difference between primary and metastatic sites (using the Fisher test of variances).

## Data Availability

The data underlying this article are available in the article and its online Appendix A.

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
