# Peer review of "Tetraploidization Increases the Motility and Invasiveness of Cancer Cells"

_ijms, 2023, doi:10.3390/ijms241813926_

Round 1

Reviewer 1 Report

Very nice work and presentation. Findings will contribute for future study designs

Author Response

# Reviewer 1

Very nice work and presentation. Findings will contribute for future study designs

We thank the reviewer for his interest in our study and positively evaluating our article.

Reviewer 2 Report

This paper highlights the role of tetraploidy in tumorigenesis and investigates its connection to metastasis, while also emphasizing the clinical significance of these findings. In this study, diploid and tetraploid clones were generated from malignant fibrous histiocytoma and colon carcinoma cells. The aim was to compare their migration and invasion capabilities in vitro and their metastatic potential in vivo. The results demonstrated that tetraploid cells had an advantage in migration and invasion in vitro, and they exhibited a greater propensity for metastasis in vivo.

Additionally, the researchers utilized the Mitelman Database of Chromosome Aberrations in Cancer to analyze the karyotypes of metastatic tumors compared to primary tumors. This analysis revealed an increased occurrence of polyploid karyotypes in metastatic tumors. This finding supports the idea that polyploidy might play a role in promoting metastasis.

The authors concluded by emphasizing the clinical importance of understanding the polyploid subpopulation's impact on patients’ survival and the need to address polyploidy in preclinical studies. The potential of targeting polyploidy as a therapeutic strategy for metastasis is highlighted as a strategic avenue for further research.

Overall, the paper effectively communicates the study's objectives, methodology, and key findings related to the association between polyploidy, metastasis, and cancer patients’ survival. It provides a clear understanding of the significance of polyploidy in cancer progression and suggests potential implications for clinical applications and further investigations.

The main advantage of this simple but elegant study is its novelty. The Authors presented new and reliable experimental data clearly indicating that early polyploidy (tetraploidy) promotes cell migration potential and decreases cell proliferation rate.

I believe this paper can be published in IJMS after a minor revision. There are only several easily corrected concerns:

1. Abstract: Please, extend the Abstract by providing a more detailed description of the methods and main findings. It would be also good to underline that tetraploidy increases invasiveness in spite of retardation of cell division rate. Also, please, underline the novelty of the study.

2. Results: The supplementary figures would look good in the main text. All of them are very informative. Also, if possible, could you provide the quantitative data showing the difference between the nucleus size (area) in diploid vs. tetraploid nuclei?

3. Please re-read the text several times to fish out small typos and inappropriate wording.

4. Discussion of the background.

The Introduction and Discussion sections are rather small and some relevant works are not mentioned.

In particular, a previous paper showed that polyploidy was significantly more common in nonsmall-cell lung, pancreatic, and prostate cancer metastases (compared with non-metastases).

(https://www.nature.com/articles/s41588-018-0165-1)

Furthermore, the authors did not outline a general background of oncogenesis (at least briefly, and in a part, which is relevant to this topic). Also, as the authors wrote, “The mechanism by which polyploid cells may acquire a preferential and enhanced migration and invasive capacity is still unknown.“ (lines 235-237)

It is pertinent to acknowledge a recent work, which addresses both the general background of oncogenesis and the role of polyplody in this process (doi.org/10.3390/ijms24076196). Firstly, this study proposed a mechanism of oncogenesis involving the atavistic reversal to a unicellular state. This transition is attributed to the higher density of protein interactions within the unicellular core of the human interactome compared to the multicellular periphery (which arose because of a gradual core-to-periphery growth of the human interactome in the evolution). Secondly, this research demonstrated that unicellular characteristics are further intensified in polyploid cancer cells as compared to diploid cancer cells.

Of particular note is the revelation that both the invasive and the polyploid cancer cells exhibit a stronger downregulation of genes associated with the cell membrane (compared with diploid cancer cells). This downregulation initiates already in diploid cancer cells (compared with normal cells). The decrease in cell membrane-related gene expression leads to a weakening of attachment with adjacent cells and extracellular matrix. This weakening can be a possible “mechanism by which polyploid cells may acquire a preferential and enhanced migration”, a phenomenon that the authors of the paper under review are looking for.

Also, it would be good to include in discussion the study outlining the direct connection between polyploidy and the induction of the main transcriptional regulators of EMT, including SNAI, TWIST, N-cadherin (DOI: 10.18632/oncotarget.12118 ).

Minor points

1. The paper would benefit from Abbreviations list (after Key words).

2. “materiel and method” --> “Materials and Methods” (line 178)

3. “fisher” --> “Fisher” (lines 198, 370)

4. “quiet” --> “quite” (line 224)

The Authors should re-read the text several times to fish out small misprints and to improve wording in some places.

Author Response

# Reviewer 2

This paper highlights the role of tetraploidy in tumorigenesis and investigates its connection to metastasis, while also emphasizing the clinical significance of these findings. In this study, diploid and tetraploid clones were generated from malignant fibrous histiocytoma and colon carcinoma cells. The aim was to compare their migration and invasion capabilities in vitro and their metastatic potential in vivo. The results demonstrated that tetraploid cells had an advantage in migration and invasion in vitro, and they exhibited a greater propensity for metastasis in vivo.

Additionally, the researchers utilized the Mitelman Database of Chromosome Aberrations in Cancer to analyze the karyotypes of metastatic tumors compared to primary tumors. This analysis revealed an increased occurrence of polyploid karyotypes in metastatic tumors. This finding supports the idea that polyploidy might play a role in promoting metastasis.

The authors concluded by emphasizing the clinical importance of understanding the polyploid subpopulation's impact on patients’ survival and the need to address polyploidy in preclinical studies. The potential of targeting polyploidy as a therapeutic strategy for metastasis is highlighted as a strategic avenue for further research.

Overall, the paper effectively communicates the study's objectives, methodology, and key findings related to the association between polyploidy, metastasis, and cancer patients’ survival. It provides a clear understanding of the significance of polyploidy in cancer progression and suggests potential implications for clinical applications and further investigations.

The main advantage of this simple but elegant study is its novelty. The Authors presented new and reliable experimental data clearly indicating that early polyploidy (tetraploidy) promotes cell migration potential and decreases cell proliferation rate.

We thank the reviewer for his deep understanding of the utility of our work

 I believe this paper can be published in IJMS after a minor revision. There are only several easily corrected concerns:

  1. Abstract: Please, extend the Abstract by providing a more detailed description of the methods and main findings. It would be also good to underline that tetraploidy increases invasiveness in spite of retardation of cell division rate. Also, please, underline the novelty of the study.

We here provide an improved version of our paper and an extended abstract. Changes are labelled in red.

  1. Results: The supplementary figures would look good in the main text. All of them are very informative. Also, if possible, could you provide the quantitative data showing the difference between the nucleus size (area) in diploid vs. tetraploid nuclei?

We thank the reviewer for his commentary regarding the quality of our work. We here added the figure showing the tetraploid cell proliferation disadvantage as main figure (Figure 2).

Regarding the nucleus area, we do apologize for the mistake; indeed, the data we are showing are nucleus area and not size (supplementary figure 2 B for example). We corrected the main figure by replacing size with area. (Figure 1D)

  1. Please re-read the text several times to fish out small typos and inappropriate wording.

We here provide an ameliorated version of our paper. Changes are labelled in red.

  1. Discussion of the background.

The Introduction and Discussion sections are rather small and some relevant works are not mentioned.

In particular, a previous paper showed that polyploidy was significantly more common in nonsmall-cell lung, pancreatic, and prostate cancer metastases (compared with non-metastases). (https://www.nature.com/articles/s41588-018-0165-1)

Furthermore, the authors did not outline a general background of oncogenesis (at least briefly, and in a part, which is relevant to this topic). Also, as the authors wrote, “The mechanism by which polyploid cells may acquire a preferential and enhanced migration and invasive capacity is still unknown.“ (lines 235-237).

It is pertinent to acknowledge a recent work, which addresses both the general background of oncogenesis and the role of polyploidy in this process (doi.org/10.3390/ijms24076196). Firstly, this study proposed a mechanism of oncogenesis involving the atavistic reversal to a unicellular state. This transition is attributed to the higher density of protein interactions within the unicellular core of the human interactome compared to the multicellular periphery (which arose because of a gradual core-to-periphery growth of the human interactome in the evolution). Secondly, this research demonstrated that unicellular characteristics are further intensified in polyploid cancer cells as compared to diploid cancer cells.

Of particular note is the revelation that both the invasive and the polyploid cancer cells exhibit a stronger downregulation of genes associated with the cell membrane (compared with diploid cancer cells). This downregulation initiates already in diploid cancer cells (compared with normal cells). The decrease in cell membrane-related gene expression leads to a weakening of attachment with adjacent cells and extracellular matrix. This weakening can be a possible “mechanism by which polyploid cells may acquire a preferential and enhanced migration”, a phenomenon that the authors of the paper under review are looking for.

Also, it would be good to include in discussion the study outlining the direct connection between polyploidy and the induction of the main transcriptional regulators of EMT, including SNAI, TWIST, N-cadherin (DOI: 10.18632/oncotarget.12118 ).

We agree with the reviewer about some limitations of our background sessions. We have provided an improved version here, with an extended introduction and discussion, taking into account the papers listed in the comments.

Minor points

  1. The paper would benefit from Abbreviations list (after Key words).

We added an abbreviations list to the manuscript, labelled in red.

  1. “materiel and method” --> “Materials and Methods” (line 178)

Changed, labelled in red.

  1. “fisher” --> “Fisher” (lines 198, 370)

Changed, labelled in red.

  1. “quiet” --> “quite” (line 224)

Changed, labelled in red.

Reviewer 3 Report

The manuscript “Tetraploidization increases the motility and invasiveness of cancer cells” describes the increased migration and invasion potential of tetraploid cancer cells when compared to diploid cells. The authors use flow cytometry and microscopy techniques to firstly identify and confirm ploidy in two cell lines, and to then demonstrate their increased motility via scratch assays and invasiveness via Boyden chamber assays. The techniques used in this manuscript are well described and relevant for the study. Overall, the manuscript presents a logical through process in tackling the question of ploidy in cancer however, there are clear writing/grammatical errors that need to be addressed. The following comments relate to the manuscript:

1. Line 85: RKO written as KRO.

2. Line 91: “p and DAPI” should read “Phalloidin and DAPI”.

3. Section 2.2 describes cells being stained with DAPI to observe the nucleus however in the figure legend for C-D it states Hoechst was used. Please clarify.

4. No where in the manuscript does the author describe what DI, D2, T1 and T2 represent. I’m assuming Diploid 1, Diploid 2, Tetraploid 1 and Tetraploid 2 pools/clones but this needs to be stated clearly early in the results section so that it is easy for the reader to understand.

5. In vitro and in vivo not consistently in italics.

6. Results section 2.2 and Figure legend 1 both mention clonogenic assays and crystal violet quantification however there are no results presented on this. Where is the crystal violet quantification results?

7. Figure Legend 1C states that a “nucleus is surrounded by a red line”. I cannot see a red line in the image.

8. Line 121: “Dig S2” should read “Fig S2”.

9. No scale bars on microscopy images in Figure 2 and Figure 3.

10. Section 2.4 states that the authors conducted in vivo demonstrating increased metastatic potential of tetraploid clones however, the data is not shown. Why have the authors omitted this data? Please include these results in either the main manuscript or in the supplementary figures section.

11. Figure 4B: graph says “Nomber” should read “Number”.

12. Line 304: “Cristal violet” should read “Crystal violet”.

Please refer to comments in above section. Moderate work is required to address grammatical errors in the manuscript so that it reads better.

Author Response

# Reviewer 3

The manuscript “Tetraploidization increases the motility and invasiveness of cancer cells” describes the increased migration and invasion potential of tetraploid cancer cells when compared to diploid cells. The authors use flow cytometry and microscopy techniques to firstly identify and confirm ploidy in two cell lines, and to then demonstrate their increased motility via scratch assays and invasiveness via Boyden chamber assays. The techniques used in this manuscript are well described and relevant for the study. Overall, the manuscript presents a logical through process in tackling the question of ploidy in cancer.

We thank the reviewer for his interest in our study

However, there are clear writing/grammatical errors that need to be addressed. The following comments relate to the manuscript:

  1. Line 85: RKO written as KRO.

Changed, labelled in red.

  1. Line 91: “p and DAPI” should read “Phalloidin and DAPI”.

Changed, labelled in red.

  1. Section 2.2 describes cells being stained with DAPI to observe the nucleus however in the figure legend for C-D it states Hoechst was used. Please clarify.

We apologise for the typo, it's actually DAPI and we've corrected it in the text.

  1. No where in the manuscript does the author describe what DI, D2, T1 and T2 represent. I’m assuming Diploid 1, Diploid 2, Tetraploid 1 and Tetraploid 2 pools/clones but this needs to be stated clearly early in the results section so that it is easy for the reader to understand.

We introduced the D1, D2, T1, T2 meaning in the revised version of the manuscript, figure legends, labelled in red

  1. In vitroand in vivo not consistently in italics.

Changed, labelled in red.

  1. Results section 2.2 and Figure legend 1 both mention clonogenic assays and crystal violet quantification however there are no results presented on this. Where is the crystal violet quantification results?

We apologise for this error and have removed the clonogenic assays from the revised version. Crystal violet quantification for proliferation assays can be found in the new figure 2

  1. Figure Legend 1C states that a “nucleus is surrounded by a red line”. I cannot see a red line in the image.

Red lines surround the nucleus, we assume that it's not clear now when the figures are included in the text, we've changed to a white line.

  1. Line 121: “Dig S2” should read “Fig S2”.

Changed, labelled in red.

  1. No scale bars on microscopy images in Figure 2 and Figure 3.

We apologize for this missing, scale bar are added to the figures

  1. Section 2.4 states that the authors conducted in vivo demonstrating increased metastatic potential of tetraploid clones however, the data is not shown. Why have the authors omitted this data? Please include these results in either the main manuscript or in the supplementary figures section.

We have added a figure showing our in vivo data, new figure 5 A-B

  1. Figure 4B: graph says “Nomber” should read “Number”.

Changed, labelled in red.

  1. Line 304: “Cristal violet” should read “Crystal violet”.

Changed, labelled in red.